# 6-Hydroxy-2,2,4-trimethyl-1,2-dihydroquinoline Demonstrates Anti-Inflammatory Properties and Reduces Oxidative Stress in Acetaminophen-Induced Liver Injury in Rats

Evgenii D. Kryl'skii [1], Svetlana E. Kravtsova [1], Tatyana N. Popova [1,*], Larisa V. Matasova [1], Khidmet S. Shikhaliev [2] and Svetlana M. Medvedeva [2]

1   Department of Medical Biochemistry, Molecular and Cell Biology, Voronezh State University, Universitetskaya sq. 1, 394018 Voronezh, Russia; krylskiy@bio.vsu.ru (E.D.K.)
2   Department of Organic Chemistry, Voronezh State University, Universitetskaya sq. 1, 394018 Voronezh, Russia
*   Correspondence: popova@bio.vsu.ru

**Abstract:** We examined the effects of 6-hydroxy-2,2,4-trimethyl-1,2-dihydroquinoline on markers of liver injury, oxidative status, and the extent of inflammatory and apoptotic processes in rats with acetaminophen-induced liver damage. The administration of acetaminophen caused the accumulation of 8-hydroxy-2-deoxyguanosine and 8-isoprostane in the liver and serum, as well as an increase in biochemiluminescence indicators. Oxidative stress resulted in the activation of pro-inflammatory cytokine and NF-κB factor mRNA synthesis and increased levels of immunoglobulin G, along with higher activities of caspase-3, caspase-8, and caspase-9. The administration of acetaminophen also resulted in the development of oxidative stress, leading to a decrease in the level of reduced glutathione and an imbalance in the function of antioxidant enzymes. This study discovered that 6-hydroxy-2,2,4-trimethyl-1,2-dihydroquinoline reduced oxidative stress by its antioxidant activity, hence reducing the level of pro-inflammatory cytokine and NF-κB mRNA, as well as decreasing the concentration of immunoglobulin G. These changes resulted in a reduction in the activity of caspase-8 and caspase-9, which are involved in the activation of ligand-induced and mitochondrial pathways of apoptosis and inhibited the effector caspase-3. In addition, 6-hydroxy-2,2,4-trimethyl-1,2-dihydroquinoline promoted the normalization of antioxidant system function in animals treated with acetaminophen. As a result, the compound being tested alleviated inflammation and apoptosis by decreasing oxidative stress, which led to improved liver marker indices and ameliorated histopathological alterations.

**Keywords:** acetaminophen; hepatotoxicity; 6-hydroxy-2,2,4-trimethyl-1,2-dihydroquinoline; oxidative stress; inflammation; caspases; antioxidant system

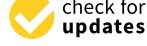



## 1. Introduction

Hepatotoxicity is the primary cause of global adverse drug reactions, often requiring the cessation of medication [1]. Paracetamol or acetaminophen (AAP) is a widely used antipyretic and analgesic. An overdose of AAP is a common cause of drug-induced liver damage and can lead to fatal acute liver injury [2]. After administering AAP, roughly 90% of it converts into non-toxic metabolites. However, only 10% undergo transformation into a reactive metabolite, namely N-acetyl-p-benzoquinone imine (NAPQI), by cytochrome P450 enzymes [3]. This metabolite is detoxified by conjugating it with glutathione (GSH). After hepatic GSH depletion, NAPQI binds to cellular proteins and causes toxicity [4,5]. The mitochondrial proteins, particularly complexes I and II of the mitochondrial electron transport chain, are the main targets of NAPQI in hepatocytes; this leads to electron leakage of oxygen and the formation of superoxide radicals. As a result, peroxynitrite may form when superoxide radicals react with endogenous nitric oxide [6]. The reaction with GSH allows for the detoxification of peroxynitrite. Therefore, the overproduction of free radicals causes additional consumption of GSH, resulting in oxidative stress [7].

Oxidative stress, induced by NAPQI, can initiate liver damage via hepatocyte death. Necrotic cell death results in the passive release of damage-associated molecular patterns (DAMPs) such as high mobility box protein 1, mitochondrial DNA, nuclear DNA fragments, histones, ATP, uric acid, and other molecules. Many of these DAMPs bind to pattern recognition receptors, such as Toll-like receptors (TLRs). These receptors are mainly found in macrophages, where they can induce transcriptional activation of cytokine and chemokine genes [8]. Nuclear factor κB (NF-κB) is a significant transcription factor involved in regulating the expression of pro-inflammatory genes, such as IL-1β, TNF-α, and cyclooxygenase-2 (COX-2). In its inactivated form, NF-κB is generally located in the cytoplasm. Upon external stimulation, it translocates to the nucleus and induces the transcription of pro-inflammatory genes. The impairment of liver regeneration is closely linked to the activation of the transcription factor NF-κB, which serves as a significant mechanism of AAP toxicity [9,10].

Cellular GSH is a redox-active reducing sulfhydryl (-SH) tripeptide found ubiquitously in the organism and is important for maintaining the redox state of the cell by directly scavenging free radicals or by participating in reactions catalyzed by antioxidant enzymes such as glutathione peroxidase (GP) or glutathione S-transferase (GST). The biosynthesis of GSH is regulated by two significant enzymes. The first is gamma-glutamylcysteinylglycine synthetase, which catalyzes the limiting step of de novo GSH biosynthesis within cells. The other enzyme is glutathione reductase (GR), which facilitates the conversion of oxidized GSH (GSSG) into the reduced form through the utilization of NADPH as a hydrogen donor [11]. NADPH is a significant metabolic product of the pentose phosphate pathway (PPP), employed to preserve catalase stability and replenish oxidized glutathione in the GR reaction. Glucose-6-phosphate dehydrogenase (G6PDH) is regarded as a crucial PPP enzyme that synthesizes NADPH and safeguards the brain against ischemic injury [12]. NADP-dependent isocitrate dehydrogenase (NADP-IDH) is an enzyme that generates NADPH and is primarily located in the cytosol and mitochondria [13]. Superoxide dismutase (SOD) is a crucial player in safeguarding cells from reactive oxygen species (ROS) and serves as the foremost defense against superoxide radicals by catalyzing their dismutation to $H_2O_2$. Catalase counters $H_2O_2$, thereby preventing the peroxidation of unsaturated fatty acids in cellular membranes. Studies reveal that reduced activity of catalase and GP can result in delayed elimination of $H_2O_2$, and excess $H_2O_2$ can, in turn, hinder the activity of catalase and GP [14]. Nuclear factor-erythroid-2-related factor 2 (Nrf2) is engaged in various cellular defense mechanisms against oxidative stress, including the detoxification of xenobiotics, as well as the regulation of cellular metabolism and participation in cell proliferation. Nrf2 has been shown to mediate the expression of cytoprotective genes through antioxidant response elements (AREs), providing regulation of a network of detoxification enzymes and enzymes involved in antioxidant metabolism. It is acknowledged that AAP can generate ROS following liver injury, thereby inducing Nrf2 activation. Furthermore, Nrf2-deficient mice exhibit greater sensitivity to AAP compared to their wild-type counterparts [15].

N-acetylcysteine remains the solitary FDA-approved remedy for AAP overdose four decades since its inception and supplements GSH to neutralize NAPQI [16]. Nevertheless, delayed use and a limited therapeutic time frame renders N-acetylcysteine relatively ineffective [17]. In addition, there are reports suggesting that prolonged administration of N-acetylcysteine disturbs normal mitochondrial processes in the liver and hinders liver regeneration after acetaminophen-induced damage by disrupting nuclear factor NF-κB signaling pathways [18]. Therefore, there is a pressing need for additional drugs for treating AAP-triggered hepatotoxicity.

Given the central role that oxidative stress plays in the pathogenesis of AAP, analyzing compounds with high antioxidant and anti-inflammatory potential seems reasonable. Dihydroquinoline derivatives can serve as such compounds, among which ethoxyquin (6-ethoxy-2,2,2,4-trimethyl-1,2-dihydroquinoline) stands out as a representative. Ethoxyquin is currently used for preventing liver diseases in pigs [19–21] and has antitoxic properties

of ethoxyquin [22]. However, the compound's use as a hepatoprotector for humans has been deemed unsuitable due to its adverse effects [23,24]. Notably, dihydroquinoline derivatives have shown more potential as drug precursors. One of the chosen substances, 6-hydroxy-2,2,2,4-trimethyl-1,2-dihydroquinoline (DHQ), underwent synthesis and evaluation as a hepato-protectant for AAP. It has been demonstrated previously that DHQ exerts hepatoprotective properties in rats with $CCl_4$-induced liver injury by regulating redox homeostasis, inhibiting apoptosis, and NLRP3 inflammasome activity [25]. The present study aimed to assess the effectiveness of DHQ as a hepatoprotective agent in rats with AAP-induced liver injury. It was also aimed to investigate the effect of DHQ on the degree of oxidative stress, inflammation, and caspase activity in rats with AAP-induced liver injury.

## 2. Materials and Methods

### 2.1. Animal Study

Male white laboratory rats of the Wistar breed, aged between 4 and 6 months and weighing between 200 and 250 g, were selected for this study. Throughout this study, the rats were assigned to four groups. Group 1 (Con, $n = 8$) comprised control animals that were kept on a standard vivarium regime and orally administered Vaseline oil. Group 2 (AAP, $n = 8$) comprised rats with AAP-induced hepatotoxicity, which was modeled by orally administering AAP dissolved in 1 mL of Vaseline oil at a dose of 1000 mg/kg of animal body weight [26]. Group 3 animals (AAP + DHQ50, $n = 8$) received oral administration of DHQ at a dose of 50 mg per 1 kg body weight, dissolved in 1 mL of 1% starch, 1 h and 12 h after paracetamol administration. Similarly, Group 4 animals (AAP + DHQ25, $n = 8$) received DHQ at a dose of 25 mg per 1 kg body weight, dissolved in 1 mL of 1% starch, 1 h and 12 h after paracetamol administration. Thus, the animals were administered a DHQ dosage equivalent to the daily dose of the widely-used hepatoprotective drug silymarin for human beings based on the weight of rats. According to guidelines, silymarin was given daily for 90 days at a dosage ranging from 105 to 210 mg, that is 135–270 mg/kg of drug per 70 kg person. In line with this, the AAP + DHQ50 group of animals received a total of 100 mg/kg BW DHQ during the entirety of the experimental period. Animals were euthanized 24 h post-treatment, and their liver and blood samples were promptly obtained for biochemical analysis.

### 2.2. Biochemical Analysis

The activity of alanine aminotransferase (ALT), aspartate aminotransferase (AST), alkaline phosphatase, total bilirubin, and cholesterol content in rat blood serum was evaluated using diagnostic kits from Olvex (St Petersburg, Russia). SOD activity was measured by the indirect method developed by Nishikimi et al. based on the reduction of nitro blue tetrazolium (NBT) [27]. In this assay, SOD competes for the superoxide radicals generated in the presence of PMS and NADH, reducing the reduction of NBT. The reaction mixture consisted of 0.33 mM EDTA, a 0.1 M phosphate buffer (pH 7.8), 0.8 mM NADH, 0.41 mM NBT, and 0.01 mM PMS. The absorbance of the formed blue formazan was measured at 540 nm. Catalase activity was assessed according to the spectrophotometric assay developed by Goth [28]. The reaction mixture contained 0.08% hydrogen peroxide in 0.1 M Tris-HCl buffer (pH 7.4) as a substrate. The decomposition of hydrogen peroxide by catalase was terminated by adding 4.5% ammonium molybdate. The intensity of the formed yellow complex was measured at 410 nm. GP activity was assayed according to Paglia and Valentina [29]. Hydrogen peroxide (0.37 mM) and GSH (0.85 mM) were used as substrates, and coupled oxidation of NADPH (0.12 mM) by GR (1 U/mL) was determined at a wavelength of 340 nm. GR activity was evaluated by measuring the oxidation of NADPH (0.16 mM) using oxidized glutathione as a substrate [30]. The reaction mixture contained a 50 mM potassium phosphate buffer (pH 7.4), 0.8 mM oxidized glutathione, 0.16 mM NADPH, and 1 mM EDTA. GST activity was assessed by the method of Warholm et al. using GSH and 1-chloro-2,4-dinitrobenzene acid as substrates [31]. The media for the

determination of G6PDH activity consisted of a 50 mM Tris-HCl buffer (pH 7.8) containing 3.2 mM glucose-6-phosphate, 0.25 mM NADP, and 1.0 mM $MgCl_2$. The activity of NADP-IDH was measured in a media consisting of 50 mM Tris-HCl buffer (pH 7.8), 1.5 mM isocitrate, 2 mM $MnCl_2$, and 0.4 mM NADP. The enzymatic activity was evaluated by the change in optical density at 340 nm using a Shimadzu UV-1900 spectrophotometer (Kyoto, Japan).

### 2.3. Assessment of Oxidative Stress Intensity

The sample's oxidative stress intensity and total antioxidant activity were measured using biochemiluminescence (BChL) induced by hydrogen peroxide with iron sulfate [32]. A BChL kinetic curve was recorded for 30 s with a BChL-07 biochemiluminometer (Medozons, Nizhny Novgorod, Russia), and the following parameters were calculated: the total light from chemiluminescence (S), maximum intensity (Imax), and tangent of the BChL kinetic curve slope (tgα2). The reaction medium consisted of 0.4 mL of 0.02 mM potassium phosphate buffer (pH 7.5), 0.4 mL of 0.01 mM $FeSO_4$, and 0.2 mL of a 2% hydrogen peroxide solution, which was introduced just before the measurement. The test material was added in a volume of 0.1 mL before the hydrogen peroxide was introduced. For the quantification of diene conjugate (DC) levels, the test sample was treated with heptane and isopropanol, mixed, and then centrifuged at $3000 \times g$ until precipitation was achieved. The heptane layer of the supernatant was subsequently diluted with ethanol and analyzed via spectrophotometry at 233 nm [33].

### 2.4. Immunoassay

Serum immunoglobulin G (IgG) levels were determined using the Rat IgG ELISA kit (Abcam, Cambridge, UK). The 8-hydroxy-2-deoxyguanosine (8-OHDG) level in rat serum and liver was determined using an 8-hydroxy 2 deoxyguanosine ELISA Kit (Abcam, Cambridge, UK), and the data were recorded on a Stat Fax 4300 chromate ELISA photometer (Awareness Technology, Palm City, FL, USA).

### 2.5. Histological Analysis

Liver hematoxylin-eosin staining was evaluated in three rats per group. Rats were anesthetized, and each section of the liver was swiftly extracted and submerged in 10% formalin for 2 h. Afterward, the liver was rinsed three times with PBS. After dehydration and embedding in paraffin, the liver tissues were sliced into 6-μm-thick sections using a Thermo Fisher Scientific HM-325 rotary microtome (Waltham, MA, USA) for hematoxylin-eosin staining. The nuclei were counterstained with hematoxylin and differentiated with eosin in the cytoplasm. Using an AxioLab A1 light microscope (Zeiss, Jena, Germany), high-magnification images were captured. A minimum of five fields were evaluated for each slide. A numerical scoring for assessing histological activity was carried out in accord with Knodell et al. [34].

### 2.6. Determination of Intensity of Apoptotic Processes

The activities of caspase-3, caspase-8, and caspase-9 enzymes were assessed using a Caspase Assay Kit (Abcam, Cambridge, UK). The kits detected caspase activation by measuring the production of p-nitroaniline, resulting from the hydrolysis of acetyl-Ile-Glu-Thr-Asp p-nitroaniline, acetyl-Asp-Glu-Val-Asp p-nitroaniline and acetyl-Leu-Glu-His-Asp p-nitroaniline by caspase-8, caspase-3, and caspase-9, correspondingly. The resulting p-nitroaniline was determined spectrophotometrically at 405 nm. The activity of caspase was expressed as the amount of product formed in picomoles per minute, per one milligram of protein.

### 2.7. Quantitative Reverse Transcription-Polymerase Chain Reaction (qRT-PCR)

Total RNA was isolated using ExtractRNA reagent (Eurogen, Moscow, Russia). The quality of RNA isolation was controlled by agarose gel electrophoresis. Reverse transcrip-

tion was performed in two repetitions using the MMLV RT kit (Eurogen, Russia) according to the instructions. Real-time PCR was performed using qPCRmix-HS SYBR mix (Evrogen, Russia) on a CFX-Connect instrument (BioRad, Hercules, CA, USA). The list of primers is provided in Table 1. The results were analyzed by the $2^{-\Delta\Delta Ct}$ method. The specificity of the reaction was evaluated by melting curves.

**Table 1.** List of primers.

| Name | Sequence |
| --- | --- |
| *Nfkb2* | F: 5′-GAATTCAGCCCCTCCATTG-3′ |
| *Nfkb2* | R: 5′-CTGAAGCCTCGCTGTTTAGG-3′ |
| *Il1b* | F: 5′-TGTGATGAAAGACGGCACAC-3′ |
| *Il1b* | R: 5′-CTTCTTCTTTGGGTATTGTTTGG-3′ |
| *Il6* | F: 5′-CCTGGAGTTTGTGAAGAACAACT-3′ |
| *Il6* | R: 5′-GGAAGTTGGGGTAGGAAGGA-3′ |
| *Tnf* | F: 5′-TCTGTGCCTCAGCCTCTTCT-3′ |
| *Tnf* | R: 5′-GGCCATGGAACTGATGAGA-3′ |
| *Ptgs2* | F: 5′-TACACCAGGGCCCTTCCT-3′ |
| *Ptgs2* | R: 5′-TCCAGAACTTCTTTTGAATCAGG-3′ |
| *Nfe2l2* | F: 5′-GCCTTGTACTTTGAAGACTGTATGC-3′ |
| *Nfe2l2* | R: 5′-GCAAGCGACTGAAATGTAGGT-3′ |
| *Gapdh* | F: 5′-CCCTCAAGATTGTCAGCAATG-3′ |
| *Gapdh* | R: 5′-AGTTGTCATGGATGACCTTGG-3′ |
| *Actb* | F: 5′-CCCGCGAGTACAACCTTCT-3′ |
| *Actb* | R: 5′-CGTCATCCATGGCGAACT-3′ |

*2.8. Statistical Analysis*

Multiple groups were analyzed using one-way ANOVA with Tukey's post hoc test; statistical significance was determined at $p < 0.05$. IBM SPSS Statistics 25 software was used for statistical analysis. Quantitative data were presented as mean $\pm$ standard deviation (SD). With a significance level of 0.05, a power of 95%, and an effect size of 2.0, a minimum of 32 animals were required for the total sample size [35]. Post hoc calculations, using a hypothetically assumed effect size and a total sample size of 32 (i.e., $n = 8$/group), indicated that the actual achieved power in this study was 95%.

**3. Results**

*3.1. DHQ Reduces AAP-Induced Hepatotoxicity in Rats*

Figure 1A illustrates increased necrotic area and loss of tissue architecture in AAP-injected rats compared to control rats. DHQ administration significantly reduced the necrotic area and alterations in liver tissue architecture in these rats. Furthermore, as indicators of clinical liver damage [36], serum levels of AST, ALT, and alkaline phosphatase were elevated after AAP treatment (Figure 1B,C). DHQ decreased the levels of these enzymes in rat serum that was injected with AAP (Figure 1B,C). A more significant effect was observed when applying a dosage of 50 mg/kg. Furthermore, the serum level of bilirubin (Figure 1D) and cholesterol (Figure 1E) increased in rats that had liver damage. Administration of DHQ decreased serum levels of these indicators of liver function in AAP-injected rats. Thus, these results indicate that DHQ has a beneficial effect on AAP-induced hepatotoxicity.

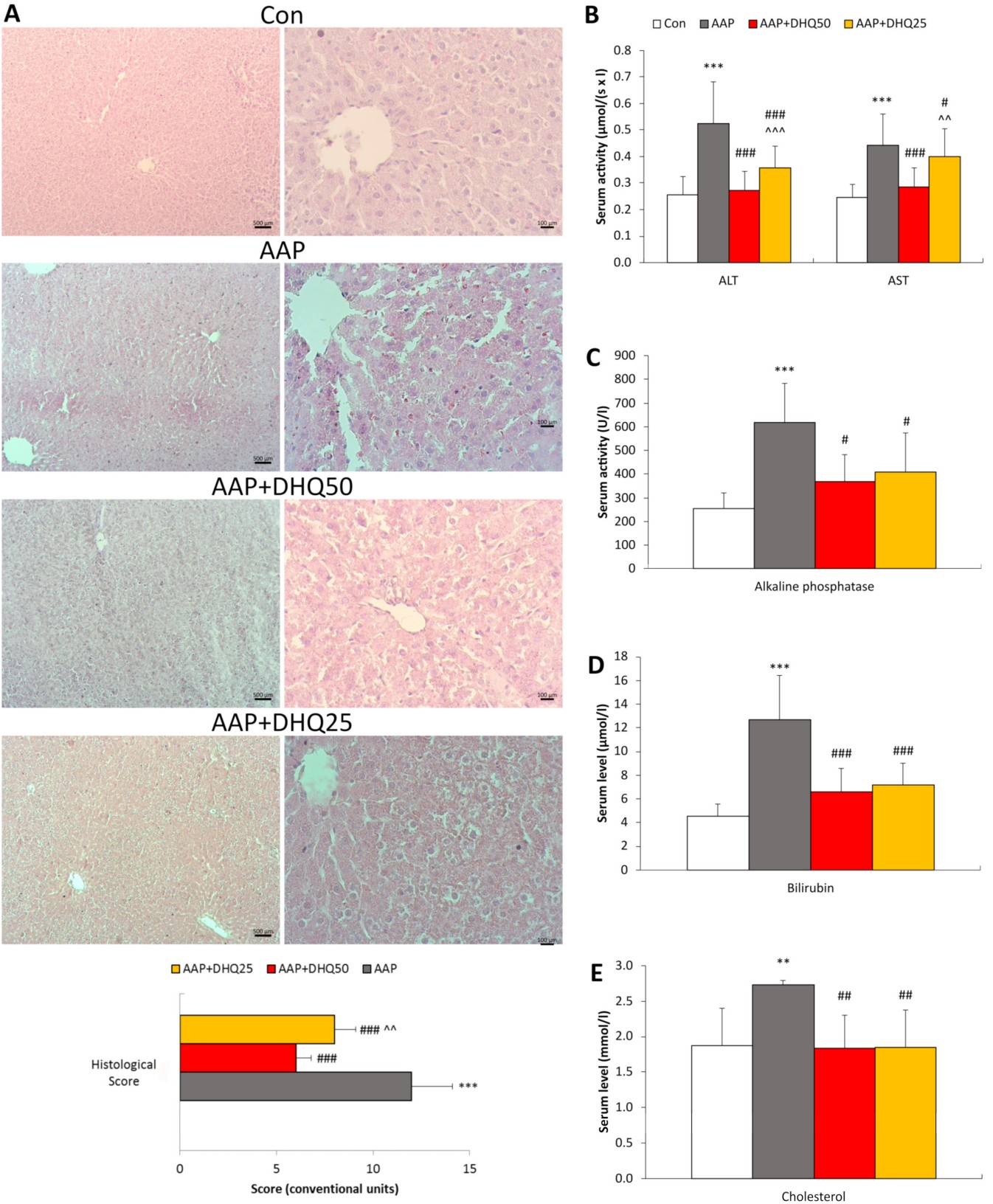

**Figure 1.** Effects of DHQ on histological abnormalities and liver index in rats with AAP-induced hepatotoxicity. (**A**) H&E staining. Scale bar: 500 μm, 100 μm. (**B**) ALT and AST activities. (**C**) Alkaline phosphatase activity. (**D**) Bilirubin level. (**E**) Cholesterol level. Data are shown as mean ± SD. ** $p < 0.01$ and *** $p < 0.001$ vs. Control. # $p < 0.05$, ## $p < 0.01$, and ### $p < 0.001$ vs. AAP. ^^ $p < 0.01$ and ^^^ $p < 0.001$ vs. AAP+DHQ50. (**A**) $n = 3$ per group; (**B**–**E**) $n = 8$ per group.

### 3.2. DHQ Reduced Oxidative Stress in Rats Injected with AAP

To evaluate oxidative stress, we measured the levels of 8-OHDG, a DNA oxidation marker [37], DC as indicators of lipid peroxidation, and BCL parameters. In rats injected with AAP, there was an increase in hepatic and serum DC concentrations, as well as hepatic and serum OHDG levels. However, DHQ reduced these parameters (Figure 2A,B). It was observed that the BCL curve parameters, Imax and S, which indicate the intensity of free radical-induced oxidation, were increased in the liver and serum of rats with paracetamol-induced liver damage compared to the control group of animals (Figure 2C,D). Furthermore, the value of $tg\alpha_2$, which characterizes the antioxidant potential, also increased compared to the control (Figure 2E), potentially as a compensatory response to amplified free radical reactions. These changes were significantly reversed by DHQ (Figure 2C–E).

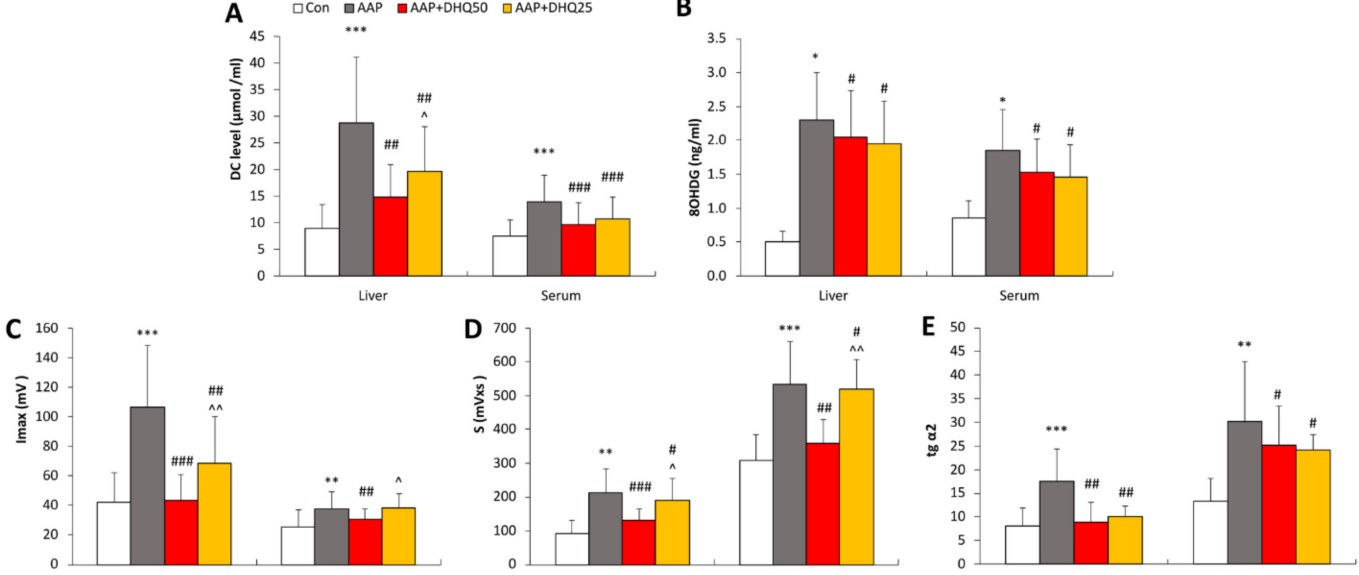

**Figure 2.** Effects of DHQ on redox status in the liver and serum of rats that received AAP. (**A**) DC levels. (**B**) 8OHDG concentration. (**C**) Imax levels. (**D**) S levels. (**E**) $tg\alpha_2$ levels. Data are shown as mean $\pm$ SD. * $p < 0.05$, ** $p < 0.01$ and *** $p < 0.001$ vs. Control. # $p < 0.05$, ## $p < 0.01$ and ### $p < 0.001$ vs. AAP. ^ $p < 0.05$ and ^^ $p < 0.01$ vs. AAP+DHQ50. $n = 8$ per group.

### 3.3. DHQ Suppresses Inflammatory Responses

Inflammatory responses play a significant role in AAP-induced hepatotoxicity [37]. Treatment with DHQ resulted in a decrease in the serum levels of IgG (Figure 3A) and in the hepatic mRNA levels of *Nfkb2*, *Il1b*, *Il6,* and *Tnf* (Figure 3B–F) compared to the AAP group. Moreover, DHQ reduces the elevated hepatic mRNA level of *Ptgs2*, which codes for cyclooxygenase-2, one of the primary enzymes involved in prostaglandin synthesis [38] (Figure 3B).

### 3.4. DHQ Decreased the Activity of Caspases in Rats with AAP-Induced Liver Injury

To estimate the intensity of apoptosis in the liver of rats, we determined the activity of caspases-3, -8, and -9 [39]. The administration of AAP increased the activity of caspases-3, -8, and -9, indicating increased apoptosis (Figure 4). DHQ reduced the activity of caspases-3, caspase-8 and caspase-9 (Figure 4).

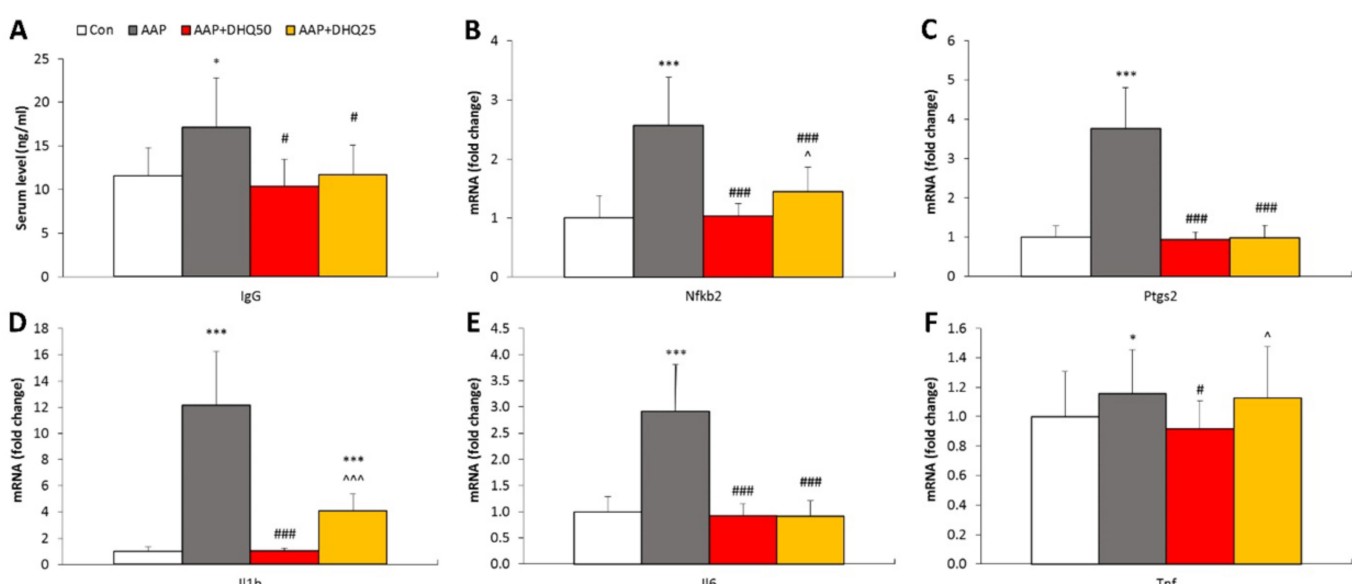

**Figure 3.** Effects of DHQ on IgG concentration and mRNA levels of inflammatory genes in rats with AAP-induced liver injury. (**A**) IgG levels. (**B**–**F**) Hepatic mRNA levels of *Nfkb2*, *Ptgs2*, *Il1b*, *Il6* and *Tnf*. Data are shown as mean $\pm$ SD. * $p < 0.05$ and *** $p < 0.001$ vs. Control. # $p < 0.05$ and ### $p < 0.001$ vs. AAP. ^ $p < 0.05$ and ^^^ $p < 0.001$ vs. AAP+DHQ50. $n = 8$ per group.

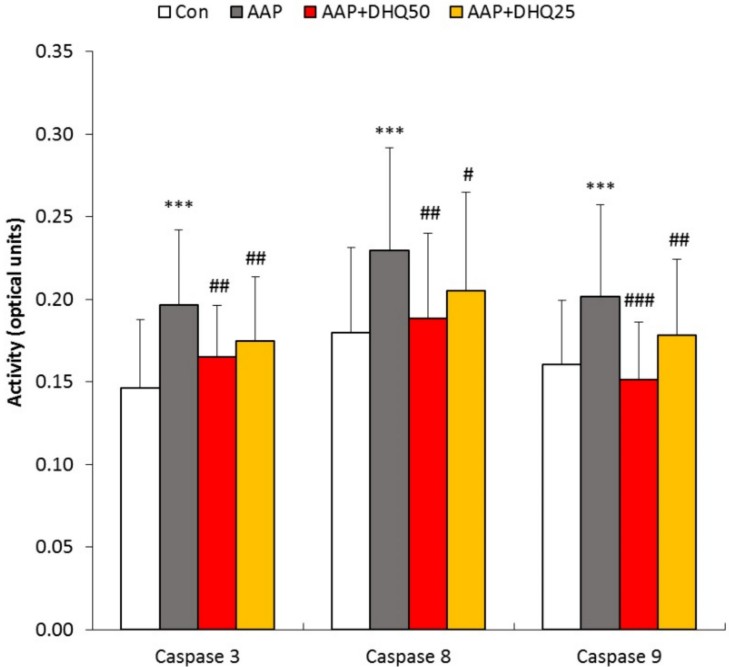

**Figure 4.** Effects of DHQ on caspase activity in rats with AAP-induced liver injury. Data are shown as mean $\pm$ SD. *** $p < 0.001$ vs. Control. # $p < 0.05$, ## $p < 0.01$ and ### $p < 0.001$ vs. AAP. $n = 8$ per group.

### 3.5. DHQ Has a Beneficial Effect on the Activity of Antioxidant Enzymes in the Liver Injury of Rats Induced by AAP

Our research indicates that the activation of free-radical oxidation following AAP administration resulted in an imbalance in the functioning of antioxidant enzymes. Therefore, a rise in the activity of SOD, catalase, and GR, alongside a decrease in the activity of GP and GST, was observed in animals from the AAP group compared with the control group (Figure 5). In addition, the concentration of GSH decreased in the liver and serum of animals with liver injury induced by AAP. Furthermore, an increase in the mRNA level

of factor Nrf2 in the liver was observed in rats with pathology. DHQ administration to rats with AAP-induced liver injury resulted in a decrease in liver SOD and catalase activity, as well as liver and serum GR, compared with the indicators in the AAP group. There was a trend towards a mild increase in SOD activity in the serum of animals in the AAP+DHQ50 and AAP+DHQ25 groups. In contrast, no significant changes were found in GP activity when DHQ was administered. However, GST activity showed an increase in the serum of rats from the AAP+DHQ50 group compared to animals with AAP-induced liver injury. Simultaneously, a substantial increase in GST activity was observed in the liver of DHQ-treated animals on an AAP background when the enzyme activity per mg of protein was recalculated (see Table 2). Additionally, the concentration of GSH in the liver and serum of AAP-treated rats was significantly higher when DHQ was administered at a dose of 50 mg/kg. The mRNA level of the Nrf2 factor was lower in the AAP+DHQ50 and AAP+DHQ25 groups than in the AAP group.

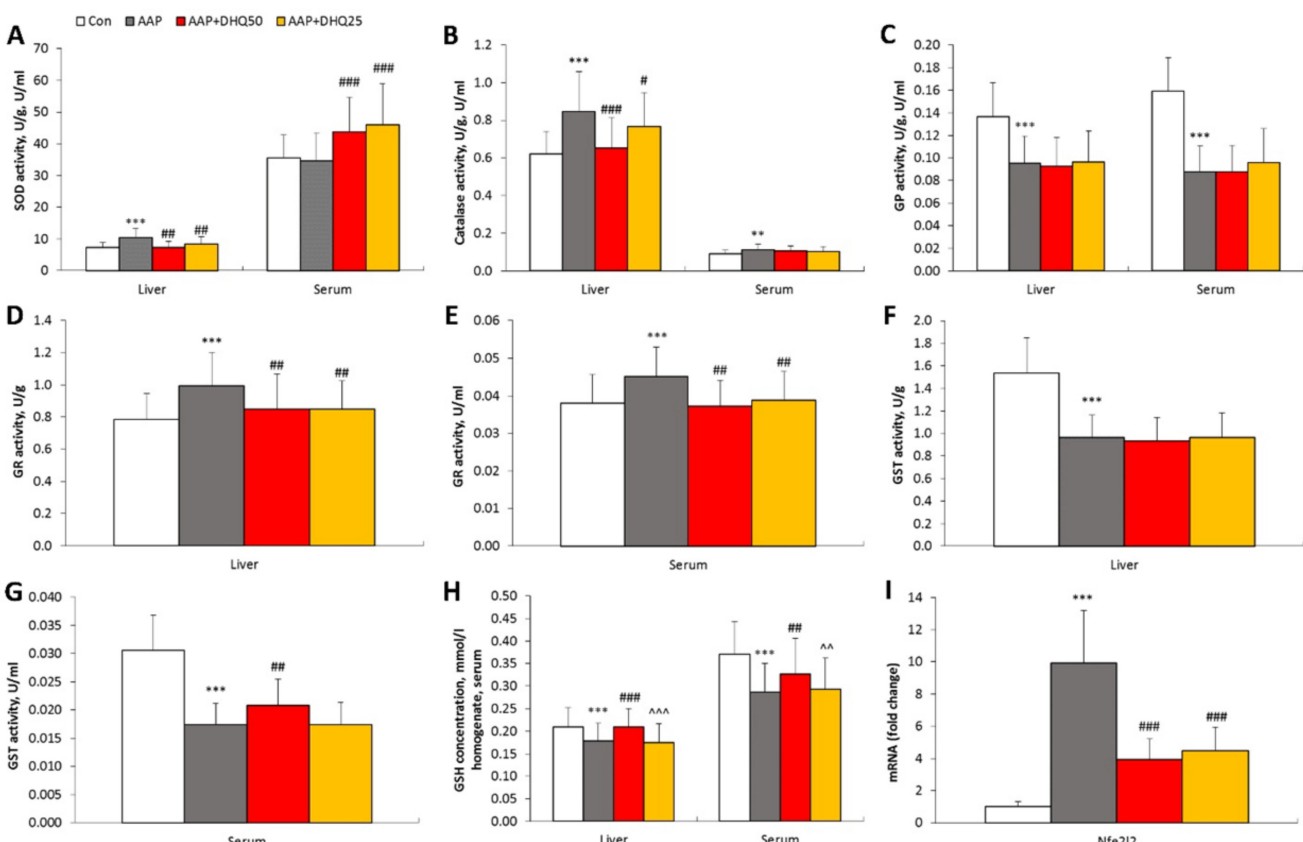

**Figure 5.** Impact of DHQ on antioxidant enzyme activity, GSH concentration, and Nrf2 (*Nfe2l2* gene) mRNA levels in rats with liver injury induced by AAP. (**A–G**) Antioxidant enzyme activity. (**H**) GSH concentration. (**I**) Hepatic mRNA level of *Nfe2l2*. Data are shown as mean $\pm$ SD. ** $p < 0.01$ and *** $p < 0.001$ vs. Control. # $p < 0.05$, ## $p < 0.01$ and ### $p < 0.001$ vs. AAP. ^^ $p < 0.01$ and ^^^ $p < 0.001$ vs. AAP+DHQ50. $n = 8$ per group.

### 3.6. DHQ Promotes Normalization of NADPH-Generating Enzymes Activity in AAP-Induced Liver Injury in Rats

Our study demonstrates that the development of AAP-induced liver injury in rats resulted in an increase in the activity of NADPH-generating enzymes-NADPH-IDH and G6PDH (Figure 6). DHQ administration led to a decrease in the activity of NADPH-IDH in the liver and G6PDH in both the liver and serum, compared to the levels found in animals from the AAP group. The enzymes' specific activity is displayed in Table 2.

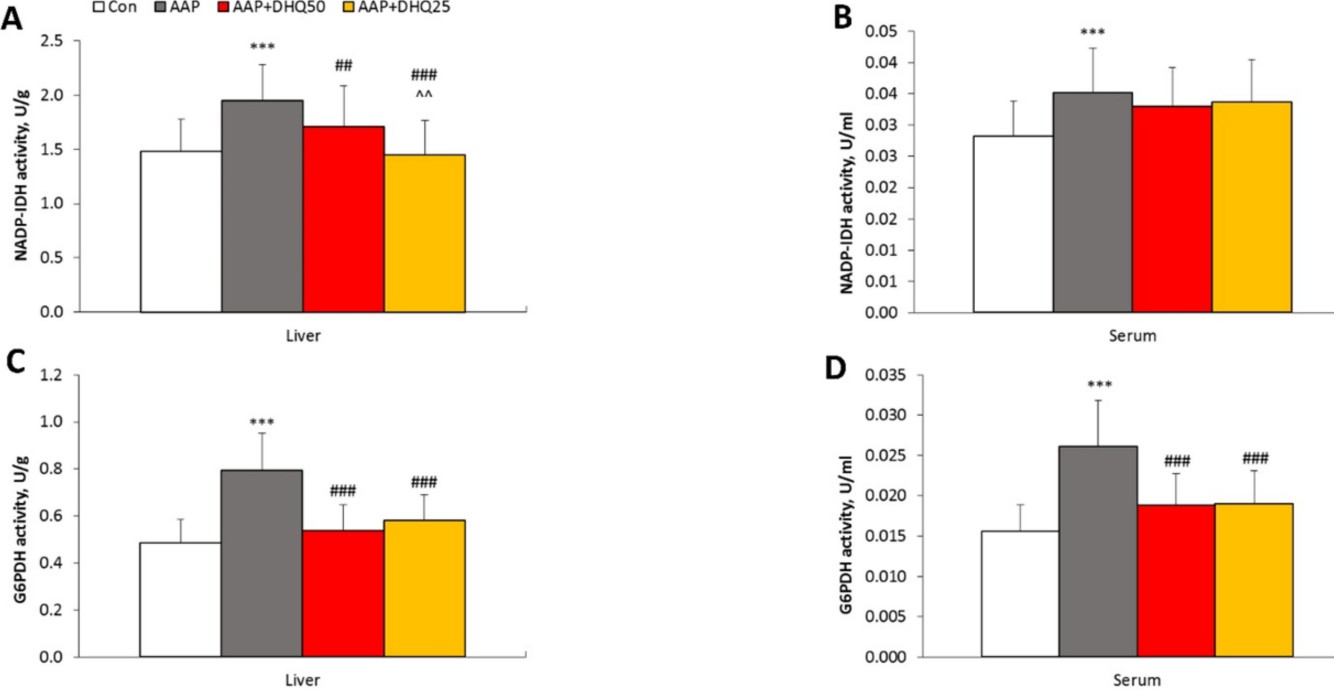

**Figure 6.** The effect of DHQ on the activity of NADPH-generating enzymes in rats with AAP-induced liver injury. (**A**,**B**) NADP-IDH activity. (**C**,**D**) G6PDH activity. Data are shown as mean ± SD. *** $p < 0.001$ vs. Control. ## $p < 0.01$ and ### $p < 0.001$ vs. AAP. ^^ $p < 0.01$ vs. AAP+DHQ50. $n = 8$ per group.

**Table 2.** Antioxidant and NADPH-generating enzymes activities, expressed in U/mg protein, in DHQ-treated rats with AAP-induced liver injury. Data are shown as mean ± SD. * $p < 0.05$ *** $p < 0.001$ vs. Control. # $p < 0.05$, ## $p < 0.01$ and ### $p < 0.001$ vs. AAP. ^^ $p < 0.01$ and ^^^ $p < 0.001$ vs. AAP+DHQ50. $n = 8$ per group.

| Indicator | Liver | | | | Blood Serum | | | |
|---|---|---|---|---|---|---|---|---|
| | Con | AAP | AAP+DHQ50 | AAP+DHQ25 | Con | AAP | AAP+DHQ50 | AAP+DHQ25 |
| SOD | 1.24 ± 0.25 | 1.62 ± 0.42 *** | 1.42 ± 0.34 # | 1.63 ± 0.42 ^^^ | 0.41 ± 0.08 | 0.45 ± 0.12 | 0.57 ± 0.16 ^^^ | 0.63 ± 0.16 ^^^ |
| Catalase | 0.0403 ± 0.0114 | 0.0491 ± 0.0123 *** | 0.0442 ± 0.0118 # | 0.0415 ± 0.0106 ## | 0.0074 ± 0.0015 | 0.0083 ± 0.0023 * | 0.0089 ± 0.0026 | 0.0086 ± 0.0024 |
| GP | 0.0118 ± 0.0024 | 0.0088 ± 0.0023 *** | 0.0088 ± 0.0027 | 0.0084 ± 0.0021 | 0.0180 ± 0.0037 | 0.0096 ± 0.0022 *** | 0.0102 ± 0.0021 | 0.0109 ± 0.0024 |
| GR | 0.0682 ± 0.0011 | 0.0828 ± 0.0172 *** | 0.0705 ± 0.0155 ### | 0.0738 ± 0.0149 ## | 0.0019 ± 0.0004 | 0.0025 ± 0.0006 *** | 0.0019 ± 0.0003 ### | 0.0020 ± 0.0005 ### |
| GST | 0.0101 ± 0.0020 | 0.0044 ± 0.0009 *** | 0.0057 ± 0.0011 ## | 0.0051 ± 0.0011 # | 0.0007 ± 0.0001 | 0.0005 ± 0.0001 *** | 0.0007 ± 0.0002 ### | 0.0006 ± 0.0001 ## |
| NADP-IDH | 0.1020 ± 0.0204 | 0.1607 ± 0.0347 *** | 0.1451 ± 0.0308 ## | 0.1381 ± 0.0328 ## | 0.0011 ± 0.0002 | 0.0015 ± 0.0004 | 0.0014 ± 0.0003 | 0.0014 ± 0.0004 |
| G6PDH | 0.0372 ± 0.0070 | 0.0663 ± 0.0142 *** | 0.0415 ± 0.0091 ### | 0.0435 ± 0.0098 ### | 0.0011 ± 0.0002 | 0.0014 ± 0.0002 *** | 0.0010 ± 0.0002 ### | 0.0012 ± 0.0002 ### ^^ |

## 4. Discussion

In this study, we performed a preliminary analysis by histological examination and quantification of serum levels of liver enzymes and functional indicators to evaluate the effect of DHQ on AAP-induced liver injury. H&E staining revealed an increase in necrosis in AAP-treated rats in comparison to control rats. Serum levels of liver enzymes were elevated after AAP injection, indicating increased permeability of hepatocyte plasma membranes and release of intracellular enzymes into the circulation. Administration of DHQ reduced the necrotic area and serum levels of AST, ALT, and AP, suggesting that DHQ has a protective effect against AAP-induced hepatotoxicity. A stronger effect was observed with a dose of 50 mg/kg.

Serum levels of bilirubin and cholesterol increased in rats injected with AAP, suggesting a disturbance in liver function regulating pigment and lipid metabolism. However, DHQ administration led to reduced serum levels of these liver function markers in AAP-injected rats. These findings provide evidence for DHQ's hepatoprotective effect.

AAP overdose results in GSH depletion and NAPQI accumulation. The excessive NAPQI triggers oxidative stress, leading to inflammation and hepatocyte death. DHQ is known to possess antioxidant properties [40,41], and we hypothesize that it can mitigate AAP-induced hepatotoxicity by inhibiting oxidative stress [37]. In this study, rats injected with AAP showed significant oxidative stress, as evidenced by increased levels of lipid peroxidation, DNA oxidation products, and elevated BCL parameters compared to the control group of animals. Administration of DHQ attenuated oxidative damage in AAP-injected rats. Previously, it was demonstrated that DHQ decreased the severity of oxidative stress and liver cell damage parameters in rats with CCl$_4$-induced liver injury [41]. Antioxidant properties have been demonstrated in several quinoline derivatives. Specifically, 6- and 8-hydroxy-substituted hydroquinoline, as well as 6- and 8-substituted amino derivatives of hydroquinoline, have shown significant antioxidant properties. It has been discovered that the addition of a hydroxyl group to the compound structure notably increases the peroxyl scavenging activity [42]. The 8-hydroxy derivative of quinoline displayed antioxidant activity, with an IC$_{50}$ of 614.77, when assessed with a 2,2-diphenyl-1-picrylhydrazyl (DPPH) assay. Furthermore, this compound is known for its ability to chelate metal ions with varying valences [43]. Other hydroxy derivatives of quinoline have also exhibited antioxidant activity in DPPH tests and have been shown to maintain the viability of mesenchymal stem cells under oxidative stress induced by H$_2$O$_2$ [44].

It is acknowledged that cytochrome P450 enzymes, particularly CYP2E1, play a crucial role in paracetamol metabolism, with resulting metabolites having been shown to cause severe hepatotoxicity. Paracetamol has been found to elevate mRNA expressions of CYP2E1, while fisetin administration has been demonstrated to reduce such expressions in a dose-dependent manner in liver tissue. Reduction in the amount of CYP2E1 enzyme has been correlated with a decrease in liver damage [45]. It is possible that the hepatoprotective effect of DHQ is partly linked to its capacity to hinder AAP biotransformation and NAPQI accumulation, owing to its structural likeness with AAP (Figure 7).

Although studies have shown that oxidative stress, mitochondrial damage, and cell death contribute to the pathogenesis of AAP hepatotoxicity [46], further research suggests that subsequent inflammatory responses of the immune system play a critical role in determining the severity and outcome of the disease [47]. Administering AAP induces excessive cytokine generation and extensive inflammatory cell infiltration [37]. In this study, DHQ reduced hepatic mRNA levels of cytokines and serum IgG levels in AAP-injected rats, suggesting that DHQ inhibits systemic and hepatic inflammation in AAP-induced hepatotoxicity. Furthermore, DHQ reduced hepatic mRNA levels of *Ptgs2*, which encodes the enzyme cyclooxygenase-2 involved in the synthesis of prostaglandins. Additionally, a dose-dependent decrease in hepatic mRNA levels of the precursor of nuclear factor kappa-light-chain-enhancer of activated B cells (NF-κB) was observed with DHQ treatment. The NF-κB signaling cascade is a crucial factor in the inflammatory responses provoked by AAP [48]. According to our prior research, the administration of DHQ significantly

reduced the CCl$_4$-induced activation of the NLRP3 inflammasome in rats, as demonstrated by decreased protein expression of NLRP3, cleaved caspase-1, and IL-1β [25]. In addition, DHQ decreased the activity of myeloperoxidase, brain mRNA levels of interleukins, and *Nfkb2* in cerebral ischemia/reperfusion in rats, thus inhibiting inflammation [40]. Currently, evidence suggests that compounds possessing antioxidant activity can suppress the expression of NF-κB and inhibit the inflammatory response. A case in point is the antioxidant berberine, which has been demonstrated to downregulate the mRNA and protein synthesis of pro-inflammatory cytokines and NF-κB through inhibition of oxidative stress [49]. In a CCl$_4$-induced liver injury model, tanshinol effectively prevented NF-κB activation and thus suppressed subsequent synthesis of pro-inflammatory cytokines. Tanshinol acted as an antioxidant by decreasing the level of malonic dialdehyde and restoring the activity of SOD and GP in rats [50]. Neferine exhibited similar mechanisms by enhancing oxidative status and suppressing inflammation and fibrosis in rats with liver damage induced by CCl$_4$ [51].

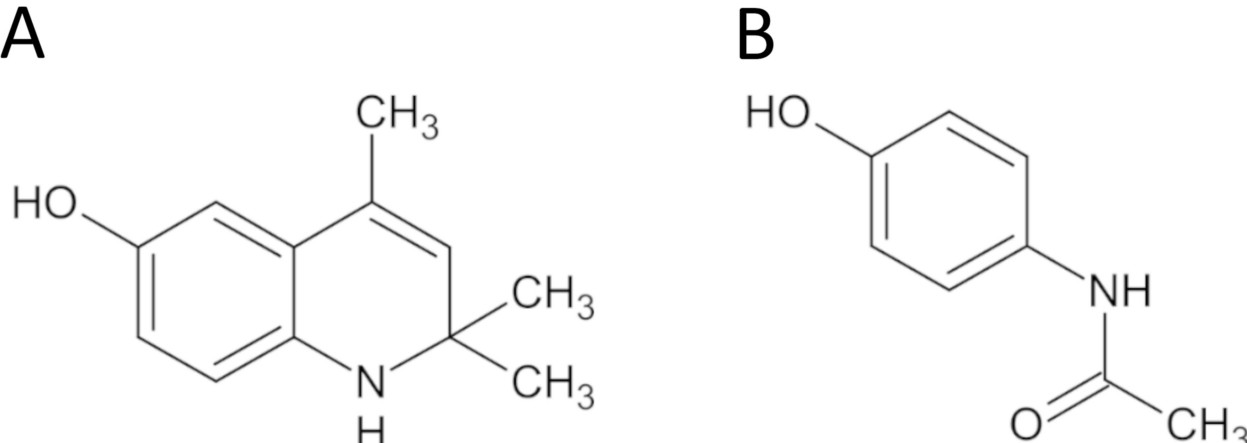

**Figure 7.** The structural formula of DHQ (**A**) and AAP (**B**).

Understanding the mechanisms of AAP-induced cell death is essential for identifying new therapeutic targets. It has been hypothesized that hepatocytes die through oncotic necrosis, apoptosis, necroptosis, ferroptosis, and pyroptosis [4]. Necrosis is widely believed to be the primary form of cell death in AAP-induced hepatotoxicity [3]. Some studies suggest that apoptosis also plays a role in AAP-triggered hepatotoxicity [37]. In order to quantify the level of apoptosis in the liver of rats, we measured the activity of caspase-3, caspase-8, and caspase-9. The inducible caspase-8 is involved in the transmission of an apoptotic signal from death ligand receptors. Activation of caspase-9 is facilitated by the assembly of the apoptosome complex at the mitochondrial apoptotic pathway; this leads to the activation of caspase-3 and caspase-7, which ultimately carry out apoptosis [39]. Caspase-3 is a significant effector caspase that cleaves multiple cytoplasmic and nuclear proteins, inducing apoptosis [52]. Our study discovered that DHQ curtailed the activity of caspase-3, caspase-8 and caspase-9. It is worth noting that apoptosis can be triggered by oxidative stress [7]. Our findings indicate that DHQ's antioxidant effect may be responsible for the inhibition of apoptosis.

The liver is protected from the effects of ROS formed during xenobiotic metabolism by the antioxidant system, comprising enzymatic and non-enzymatic components. Our research has demonstrated that the administration of paracetamol to rats disrupts antioxidant enzyme function, leading to decreased levels of GP and GST activities. The observed alterations were evidently due to the conjugation of AAP's metabolic products with GSH, a substrate for these enzymes. The collected data substantiated that the concentration of GSH was reduced in rats of the AAP group compared to control animals. Additionally, administering AAP resulted in the amplified activity of SOD and catalase, which could be a compensatory reaction of the animal organism to the excessive production of ROS. The

existing literature provides further support for our findings on the significant impact of antioxidant system imbalances in the development of xenobiotic-induced liver injury [53]. Among other findings, our study demonstrated that rats with liver injury induced by AAP showed activation of GR, as well as of the main enzymes responsible for generating NADPH for GSSH recovery, namely NADPH-IDH and G6PDH. Subsequent administration of DHQ to animals with AAP-induced liver injury resulted in increased GSH concentration and a shift in GR, NADPH-IDH, and G6PDH activity towards control, which may be indicative of a lowered requirement for NADPH for GSSH reduction. Furthermore, DHQ facilitated normalization of SOD and catalase activity in the liver of the animals and restored GST activity in AAP-treated rats, apparently as a result of inhibition of free radical-induced oxidation. It is established that Nrf2 is the primary transcription factor that regulates the antioxidant system and is sensitive to the redox state in the cell. Under the influence of ROS, modifications are made to cysteine residues within Keap1, which acts as an inhibitor for Nrf2. Such modifications prompt conformational changes in Keap1, ultimately resulting in Nrf2 ubiquitination being inhibited. As a response to this inhibition, Nrf2 is phosphorylated and undergoes translocation to the nucleus, where it binds to ARE elements [54]. The gathered data demonstrates that the administration of AAP to rats caused the accumulation of Nrf2 mRNA; this could potentially be due to the activation of Nrf2 by ROS. Simultaneously, DHQ shifted the mRNA levels of Nrf2 towards control, apparently by reducing ROS levels in the rat liver.

Thus, due to its antioxidant properties, DHQ decreased the severity of oxidative stress and apoptosis while also contributing to the normalization of the functioning of the antioxidant system. In addition, the compound efficiently suppresses hepatic and systemic inflammation. These results indicate that DHQ may be a promising novel antioxidant for the treatment of paracetamol overdose.

*Limitations*

The present study has limitations due to the narrow range of DHQ doses tested. While DHQ did significantly alter most of the parameters studied, a more extensive investigation of the hepatoprotective activity of the compound should be conducted in the future to determine the optimal concentrations and administration regimens. Another limitation of this study is the lack of an in-depth examination of DHQ's anti-inflammatory properties. It is recommended to analyze the levels of other cytokines associated with drug-induced liver damage and assess liver infiltration by various immune cells. However, the results obtained in the course of the work on the realization of the inflammatory response in conditions of AAP-induced hepatotoxicity and DHQ action are reliable and consistent with the available literature data. This study's limitations involve the absence of a group of animals receiving a comparator drug against AAP. To clarify the deeper mechanisms of DHQ's protective action and to select the optimal doses, it would be advisable to investigate its efficacy in comparison with a comparator, which could be the subject of future studies.

**Author Contributions:** Conceptualization, E.D.K., K.S.S. and T.N.P.; Methodology, E.D.K. and T.N.P.; Validation, E.D.K. and L.V.M.; Investigation, E.D.K., S.E.K. and S.M.M.; Resources, K.S.S. and T.N.P.; Data Curation, S.E.K.; Writing—L.V.M.; Writing—Review and Editing, E.D.K.; Visualization, E.D.K. All authors have read and agreed to the published version of the manuscript.

**Funding:** This research received no external funding.

**Institutional Review Board Statement:** Ethics Committee Name: Ethics Committee on Biomedical Research Expertise in Voronezh State University. Approval Code: № 42-04. Approval Date: 2 November 2020.

**Informed Consent Statement:** Not applicable.

**Data Availability Statement:** Data is contained within the article.

**Conflicts of Interest:** The authors declare no conflict of interest.

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
