# Peer review of "6-Hydroxy-2,2,4-trimethyl-1,2-dihydroquinoline Demonstrates Anti-Inflammatory Properties and Reduces Oxidative Stress in Acetaminophen-Induced Liver Injury in Rats"

_cimb, doi:10.3390/cimb45100525_

Round 1

Reviewer 1 Report

1.     The authors examined the effects of 6-hydroxy-2,2,4-trimethyl-1,2-dihydroquinoline on markers of liver injury, oxidative status, and the extent of inflammatory and apoptotic processes in rats with acetaminophen-induced liver damage. The study discovered that 6-hy-droxy-2,2,4-trimethyl-1,2-dihydroquinoline reduced oxidative stress by its antioxidant activity, hence reducing the level of pro-inflammatory cytokine and NF-κB mRNA, as well as decreasing the concentration of immunoglobulin G. So, the authors could have made a better paper if they had emphasized more in the discussion the new findings of this study.

2.     Figure 1A, The shades of HE staining are so different that it is difficult to determine the degree of liver damage. Both weakly and strongly magnified images should be shown.

3.     All figure legends should indicate the sample size.

4.     3.3. DHQ suppresses inflammatory responses, Leukocyte infiltration of the liver should be analyzed. For example, neutrophils, macrophages, lymphocytes, etc. by immunostaining.

5.     Has the gene or protein expression of IFN-g been analyzed? IFN-g is an aggravating factor in APAP liver injury (FASEB J. 2002 Aug;16(10):1227-36.).

Author Response

Dear Editor and Reviewers!

We appreciate your efforts to review our article, which provided us with valuable feedback on how to enhance the quality of the manuscript and plan for future work. The revisions made to the text according to the comments have been highlighted for your convenience. We present our responses to the reviewers' comments below.

Best regards, the authors.

Reviewer 1

  1. The authors examined the effects of 6-hydroxy-2,2,4-trimethyl-1,2-dihydroquinoline on markers of liver injury, oxidative status, and the extent of inflammatory and apoptotic processes in rats with acetaminophen-induced liver damage. The study discovered that 6-hy-droxy-2,2,4-trimethyl-1,2-dihydroquinoline reduced oxidative stress by its antioxidant activity, hence reducing the level of pro-inflammatory cytokine and NF-κB mRNA, as well as decreasing the concentration of immunoglobulin G. So, the authors could have made a better paper if they had emphasized more in the discussion the new findings of this study.

We have extended the discussion concerning the antioxidant and anti-inflammatory activity of DHQ.

  1. Figure 1A, The shades of HE staining are so different that it is difficult to determine the degree of liver damage. Both weakly and strongly magnified images should be shown.

High magnification images have been added to the article. The colour tones of the images have also been aligned.

  1. All figure legends should indicate the sample size.

The sample size in the groups indicated.

  1. 3.3. DHQ suppresses inflammatory responses, Leukocyte infiltration of the liver should be analyzed. For example, neutrophils, macrophages, lymphocytes, etc. by immunostaining.

Undoubtedly, an examination of liver infiltration by immune system cells and an investigation into the impact of DHQ on the level and transcription of other humoral agents of immune response, specifically IFN-g, would enable a more thorough evaluation of the compound's anti-inflammatory mechanisms. The present study aims to investigate the hepatoprotective properties of DHQ by examining its antioxidant effects and ability to maintain proper functioning of the antioxidant system in drug-induced liver damage. This manuscript proposes the primary mechanisms of action of DHQ, and their association with alterations in oxidative stress severity. An extensive investigation into the regulatory impact of DHQ on the immune system's efficacy is contemplated as a forthcoming research endeavour. We also assessed the extent of histopathological alterations following Knodell et al. [Knodell, R.G.; Ishak, K.G.; Black, W.C.; Chen, T.S.; Craig, R.; Kaplowitz, N.; Kiernan, T.W.; Wollman, J. Formulation and Application of a Numerical Scoring System for Assessing Histological Activity in Asymptomatic Chronic Active Hepatitis. Hepatology 1981, 1, 431-435]. The diagram in Figure 1 illustrates these findings.

  1. Has the gene or protein expression of IFN-g been analyzed? IFN-g is an aggravating factor in APAP liver injury (FASEB J. 2002 Aug;16(10):1227-36.).

Indeed, as demonstrated by this study, IFN-g is a promising therapeutic target for paracetamol-induced liver damage. The effects of DHQ on IFN-g mRNA and protein synthesis, as well as on the pathways regulating its expression in T cells, undoubtedly warrant special attention. However, the current focus of this study is to elucidate the general mechanisms of hepatoprotective action of DHQ. The chosen pro-inflammatory cytokine mRNA targets for this study are the most commonly utilised for analysing inflammatory response levels in liver pathology [https://doi.org/10.1016/j.biopha.2019.108704; https://doi.org/10.1371/journal.pone.0107405; https://doi.org/10.1039/D0FO02894K; https://doi.org/10.3390/molecules22101781; https://doi.org/10.1155/2019/9056845]. However, further extensive research is required to scrutinise the inflammatory response aspects in the context of DHQ effects which will be explored in subsequent works.

Reviewer 2 Report

1. Please state more details about 6-hydroxy-2,2,4-trimethyl-1,2-dihydroquinoline in the introduction section. It is totally inadequate in the present version.

2. What is the rationale of choosing 25 and 50 mg/kg dose? Justification should be provided.

3. It is also a limitation that a wide variety of dosing ranges were not tested for 6-hydroxy-2,2,4-trimethyl-1,2-dihydroquinoline in the present study.

4. Please provide ranges of the post-hoc power analyses for the outcomes evaluated in the present study.

5. Please state the limitations of the study separately.

Author Response

Dear Editor and Reviewers!

We appreciate your efforts to review our article, which provided us with valuable feedback on how to enhance the quality of the manuscript and plan for future work. The revisions made to the text according to the comments have been highlighted for your convenience. We present our responses to the reviewers' comments below.

Best regards, the authors.

Reviewer 2

  1. Please state more details about 6-hydroxy-2,2,4-trimethyl-1,2-dihydroquinoline in the introduction section. It is totally inadequate in the present version.

We have expanded the description of dihydroquinoline derivatives in the Introduction.

  1. What is the rationale of choosing 25 and 50 mg/kg dose? Justification should be provided.

A description of the dosage calculation has been added to the text.

  1. It is also a limitation that a wide variety of dosing ranges were not tested for 6-hydroxy-2,2,4-trimethyl-1,2-dihydroquinoline in the present study.

This item has been added to Limitations.

  1. Please provide ranges of the post-hoc power analyses for the outcomes evaluated in the present study.

Post-hoc power analyses have been added to the Statistical Analyses section

  1. Please state the limitations of the study separately.

The section with study limitations was added to the manuscript.

Reviewer 3 Report

The manuscript titled "6-Hydroxy-2,2,4-trimethyl-1,2-dihydroquinoline demonstrates anti-inflammatory properties and reduces oxidative stress in acetaminophen-induced liver injury in rats" presents a captivating investigation into the potential therapeutic effects of 6-hydroxy-2,2,4-trimethyl-1,2-dihydroquinoline on liver injury induced by acetaminophen in rats. This study delves into a highly relevant and significant issue concerning drug-induced liver damage, which has substantial clinical implications. The study's results are quite promising, suggesting that 6-hydroxy-2,2,4-trimethyl-1,2-dihydroquinoline may offer protection against acetaminophen-induced liver damage. Particularly noteworthy are its abilities to reduce oxidative stress, modulate pro-inflammatory cytokine levels, and inhibit apoptosis-related caspases. Furthermore, the restoration of the antioxidant system's function is a crucial aspect of liver protection. While the work is certainly commendable, I have a few suggestions for the authors:

1.     It has been reported that mitochondrial dysfunction plays a vital role in acetaminophen overdose-induced organ injury. Therefore, it would be worthwhile to measure mitochondrial polarization levels in this model. Additionally, exploring the impact of DHQ on mitochondrial polarization could provide valuable insights. Considering that PPARg plays a role in controlling oxidative metabolism and regulating mitochondrial function, this aspect should also be investigated.

2.     In this study, the authors measured the mRNA levels of several pro-inflammatory cytokines. While the current conclusions are based on these results, it would be beneficial to further explore the molecular function of DHQ. To address this question comprehensively, I suggest considering RNA sequencing, which could provide a more in-depth understanding of DHQ's mechanisms of action.

Author Response

Dear Editor and Reviewers!

We appreciate your efforts to review our article, which provided us with valuable feedback on how to enhance the quality of the manuscript and plan for future work. The revisions made to the text according to the comments have been highlighted for your convenience. We present our responses to the reviewers' comments below.

Best regards, the authors.

Reviewer 3

The manuscript titled "6-Hydroxy-2,2,4-trimethyl-1,2-dihydroquinoline demonstrates anti-inflammatory properties and reduces oxidative stress in acetaminophen-induced liver injury in rats" presents a captivating investigation into the potential therapeutic effects of 6-hydroxy-2,2,4-trimethyl-1,2-dihydroquinoline on liver injury induced by acetaminophen in rats. This study delves into a highly relevant and significant issue concerning drug-induced liver damage, which has substantial clinical implications. The study's results are quite promising, suggesting that 6-hydroxy-2,2,4-trimethyl-1,2-dihydroquinoline may offer protection against acetaminophen-induced liver damage. Particularly noteworthy are its abilities to reduce oxidative stress, modulate pro-inflammatory cytokine levels, and inhibit apoptosis-related caspases. Furthermore, the restoration of the antioxidant system's function is a crucial aspect of liver protection. While the work is certainly commendable, I have a few suggestions for the authors:

  1. It has been reported that mitochondrial dysfunction plays a vital role in acetaminophen overdose-induced organ injury. Therefore, it would be worthwhile to measure mitochondrial polarization levels in this model. Additionally, exploring the impact of DHQ on mitochondrial polarization could provide valuable insights. Considering that PPARg plays a role in controlling oxidative metabolism and regulating mitochondrial function, this aspect should also be investigated.
  2. In this study, the authors measured the mRNA levels of several pro-inflammatory cytokines. While the current conclusions are based on these results, it would be beneficial to further explore the molecular function of DHQ. To address this question comprehensively, I suggest considering RNA sequencing, which could provide a more in-depth understanding of DHQ's mechanisms of action.

Undoubtedly, the investigation of fundamental mechanisms responsible for the pathogenesis of toxic liver injury, such as mitochondrial dysfunction and disturbance of mitochondrial biogenesis mechanisms, merits special attention. Implementing interventions that utilise antioxidants with protective effects on such organelles appears to be a promising avenue for the development of liver injury therapy. The current study aimed to investigate the hepatoprotective properties of DHQ by examining its antioxidant action and its ability to sustain normal functioning of the antioxidant system in drug-induced liver injury. This paper proposes the underlying molecular mechanisms of DHQ action and their association with changes in the severity of oxidative stress. As a topic for future research, we propose an extensive examination of DHQ regulatory impacts on mitochondrial homeostasis in paracetamol-induced liver injury. We are planning a comprehensive investigation into the mechanisms of DHQ anti-inflammatory effects, which will be a separate research project. The study will involve evaluating changes in mRNA and protein levels of various components of the humoral response of the immune system, as well as assessing the state of cellular immunity. RNA sequencing of the inflammatory response genes that are regulated by DHQ will be a valuable addition to future work and enhance our understanding of the mechanisms underlying the anti-inflammatory actions of DHQ.

Round 2

Reviewer 1 Report

Accept in present form

Reviewer 2 Report

Thanks.